# Indoor Environmental Quality (IEQ) and Sustainable Development Goals (SDGs): Technological Advances, Impacts and Challenges in the Management of Healthy and Sustainable Environments

Iasmin Lourenço Niza * , Ana Maria Bueno  and Evandro Eduardo Broday 

IEQ Lab, Federal University of Technology—Paraná (UTFPR), Rua Doutor Washington Subtil Chueire 330, Jardim Carvalho, Ponta Grossa 84017-220, Brazil; anam@utfpr.edu.br (A.M.B.); broday@utfpr.edu.br (E.E.B.)
* Correspondence: niza@alunos.utfpr.edu.br

**Abstract:** The growing concern for sustainability is evident, given the importance of guaranteeing resources for the next generations, especially in the face of increasing energy consumption in buildings. Regardless of the context, people seek comfort, which makes investigating Indoor Environmental Quality crucial. This covers aspects such as indoor air, temperature, noise and lighting, positively impacting quality of life, reducing stress, saving energy and promoting health, well-being and productivity. A literature review was conducted using the Scopus and PubMed databases to analyze technological advances and challenges in managing healthy and sustainable environments, focusing on the relationship between Indoor Environmental Quality and the Sustainable Development Goals. Initially, 855 articles were identified, of which 123 were selected based on established criteria. Three research questions (RQs) were formulated, leading to the following conclusions. (i) The assessment of sustainability in buildings is crucial, encompassing economic, social and environmental aspects. Furthermore, the COVID-19 pandemic has underscored the importance of adapting energy strategies, thereby contributing to the achievement of the Sustainable Development Goals through the utilization of advanced technologies that promote healthy and efficient environments. (ii) Evaluations have evolved, ranging from energy savings to human well-being and mental health, including disease prevention strategies. (iii) Challenges in managing the promotion of Indoor Environmental Quality include excessive resource consumption, emissions and economic–environmental balance.

**Keywords:** indoor environmental quality; sustainable development goals; sustainability; environmental management; urban environments; sustainable construction

## 1. Introduction

The term "sustainable development" began to be widely used in 1987, when the World Commission on Environment and Development, also known as the Brundtland Commission, introduced this concept. This concept addresses the interaction between the environment and the promotion of development, ensuring current needs without compromising future generations [1]. However, concerns about this issue date back to the 18th century, coinciding with the advent of the First Industrial Revolution, when the increased use of machines to the detriment of human power demanded the use of fossil fuels for energy, resulting in more significant greenhouse gas emissions and global warming problems [2].

Today, construction is also a significant concern, as it contributes significantly to society in terms of infrastructure and consumer products. Therefore, projects in this field must incorporate sustainable practices to mitigate environmental impacts [3] and integrate social, economic and environmental issues into the development process [4]. In this way, assessing the energy performance of buildings is of fundamental importance

when pursuing sustainable development [5]. For example, energy-inefficient buildings that lack waste-sorting and water-reuse systems can lead to higher operating costs [6].

In the same context, building cooling and ventilation strategies have undergone notable transformations. Before the 1950s, ventilation and air-conditioning systems were considered economically unviable, resulting in a greater prevalence of natural ventilation to maintain thermal comfort in buildings, mainly due to the availability of windows that provide access to fresh air and natural light [7]. However, with technological advances and the growth of urban areas, architecture has evolved, resulting in the more widespread adoption of mechanical cooling and ventilation systems. At the same time, operable windows have been replaced by fully glazed structures with no possibility of opening, rendering previous designs obsolete. Accordingly, the concept of bioclimatic architecture emerges, where the construction is planned in a way that is adapted to the local climate and employs passive approaches to enhance the quality of the indoor environment, aiming to minimize energy consumption to the maximum degree [8]. As stated by Xhexhi [9], a bioclimatic project can optimize the natural environment by utilizing renewable energy systems and clean energy, reducing energy consumption for heating, cooling, and lighting and improving the quality of life in urban areas.

According to Deslatte et al. [10], governments in different nations are exploring methods to foster sustainability. For example, in the United States, investments have been directed toward renovating "green" or energy-efficient homes, addressing social inequalities and climate change, and promoting economic and infrastructural recovery. Lawrence et al. [11] point out that the contemporary challenge lies in balancing energy supply and inhabitant comfort, requiring strategies that combine Indoor Environmental Quality (IEQ) with energy consumption.

The American Society of Heating, Refrigeration, and Air-Conditioning Engineers (ASHRAE) [12] defines Indoor Environmental Quality (IEQ) as the internal perception that encompasses building planning, evaluation, and operation elements that prioritize energy efficiency, health, and comfort. In a recent study, Lolli, Coruzzolo, and Balugani [13] emphasize the importance of assessing IEQ in academic and project contexts. Therefore, direct weighting approaches can be applied to assess the risks affecting the overall environmental evaluation based on feedback provided by indoor space users, such as the Analytic Hierarchy Process (AHP) [14]. This is a characterization of Post-Occupancy Evaluation (POE), which aims to identify differences between the design stage and the current user experience in terms of design, construction, management, or misuse of the building [15].

In addition, there has been growing concern about sustainability in the built environment, focusing on mitigating environmental impacts and promoting social benefits [16]. This concern has led to the popularization of the term "energy poverty", especially to describe the situation in which many families lack the material and social resources necessary to provide adequate energy and indoor comfort to their members. This negatively impacts their quality of life, as their heating needs are not met [17]. The influence of climate change on buildings reinforces the need for approaches that expand energy efficiency and sustainable development, incorporating renewable energy sources and conscious consumption [18].

However, the pursuit of energy efficiency and occupant comfort are not always aligned. Problems associated with energy consumption and IEQ in buildings have been pointed out by several studies [19], covering various aspects that influence people's satisfaction with the built environment, such as air quality, thermal comfort, acoustics, lighting [20], furniture, maintenance, cleanliness, vibration, technology, aesthetics, appearance, privacy, and views, among others, which will serve to investigate health and productivity, thus making it possible to pay greater attention to sustainability research [21] along with the progress of the renewable energy sector, which presents numerous technological innovations in both industry and research laboratories [22]. It is worth noting that adopting technologies does not always result in a decrease in energy consumption. Therefore, projects should increasingly consider the behaviors, needs, and preferences of occupants to enhance the

indoor environment of buildings [23]. From this perspective, adopting technologies involves using new methods, systems, devices, or practices aimed at improving performance, energy efficiency, sustainability, and other aspects related to the quality and effectiveness of buildings.

In this context, the Sustainable Development Goals (SDGs), which were launched by the UN in 2015, play a key role in setting goals to be achieved by 2030 [24]. Consisting of 17 goals, 169 targets and 231 indicators, the SDGs broadly cover economic, social, and environmental issues [25]. Responding to these ambitious goals, research exploring the relationship between Indoor Environmental Quality (IEQ) and sustainable development is on the rise, in line with the purposes of the SDGs. Studies such as that by Sperry and Bender [26] address the connection between sustainability, the use of building materials and water and energy consumption. Calvo et al. [27] explore the implementation of the Internet of Things (IoT) to monitor IEQ and reduce energy consumption. Attaianese et al. [28] focus on sustainable design and ergonomics. Atanda and Olukoya [29] highlight using low-cost materials to promote sustainability in housing programs.

Within this context, this work aims to analyze technological advances and challenges in managing healthy and sustainable environments, focusing on the relationship between Indoor Environmental Quality (IEQ) and the Sustainable Development Goals (SDGs) through a literature review. This approach seeks to fill the knowledge gap concerning assessing the impacts of Indoor Environmental Quality (IEQ) on fulfilling the Sustainable Development Goals (SDGs) in built environments. This knowledge gap lies in the lack of understanding of how IEQ influences the achievement of the Sustainable Development Goals (SDGs), which are global targets established by the United Nations to promote sustainable development. This lack of understanding extends to issues such as the adoption of technologies and methodologies to effectively manage IEQ, the assessment of environmental impacts and risks when promoting healthy and sustainable conditions in both public and private environments, and the challenges and opportunities associated with urban environmental management, considering the balance between sustainable development and environmental preservation.

To address this gap, the present study proposes three research questions (RQs):

- **RQ1:** How can adopting technologies and methodologies contribute to efficiently managing Indoor Environmental Quality (IEQ) in environments, promoting healthy and sustainable conditions?
- **RQ2:** How can the assessment of environmental impacts and risks be applied to guarantee Indoor Environmental Quality (IEQ) in public and private environments when following the principles of sustainable development?
- **RQ3:** What are the main challenges and opportunities in environmental management and policy to promote Indoor Environmental Quality (IEQ) in urban environments, considering the balance between sustainable development and environmental preservation?

In summary, this study aims to fill the knowledge gap by examining how IEQ influences the attainment of the SDGs by using the research questions to investigate specific aspects related to management, assessment, and environmental policies in built environments. The goal is to contribute to developing strategies that promote buildings aligned with the purposes of the SDGs, providing comfort to occupants and preserving resources for future generations through responsible consumption.

## 2. Materials and Methods

*PRISMA Methodology, Search Strategy and Software Used*

To develop the systematic literature review, rigorous procedures have been established to identify the most relevant articles to address the three research questions (RQs). The Preferred Reporting Items for Systematic Reviews and Meta-Analyses (PRISMA), as developed by Moher et al. [30], was adapted for this study. PRISMA uses combinations of specific keywords to enable efficient searching of scientific databases. Over the last

few years, several studies have used this method to conduct literature reviews on subjects such as the association between Indoor Environmental Quality, individual productivity and organizational performance [31] and flexible learning environments [32]. In addition, the PRISMA methodology has also been applied in reviews related to the Sustainable Development Goals, exploring the relationship with blockchain technology [33] and the forestry sector [34].

The search strategy was carried out using combinations of keywords together with Boolean operators in the Scopus and PubMed databases: (("Indoor Environmental Quality" OR "IEQ") AND ("Sustainability" OR "SDG" OR "Sustainable Development Goals" OR "environmental technologies" OR "environmental methodologies" OR "environmental management" OR "environmental policy" OR "environmental risks" OR "environmental impacts")). The choice of the Scopus database was based on the relevance and scope demonstrated by Falagas et al. [35], making it suitable for this review. The search was conducted on 8 September 2023, and inclusion and exclusion criteria were applied, as shown in Table 1. These criteria were used to select the articles suitable for the research, considering the keywords, abstracts and titles of papers published over the years.

**Table 1.** Inclusion and exclusion criteria.

| Inclusion Criteria | Exclusion Criteria |
| --- | --- |
| Articles in English | Articles in other languages |
| Articles published over the years, with no time limit | Articles with missing data |
| Articles published in journals | Duplicate articles, conference papers, book chapters, conference reviews, books, editorials or short papers |
| Articles with a link between Indoor Environmental Quality (IEQ) and the Sustainable Development Goals (SDGs) | Articles that do not align with the research theme |
| Articles capable of addressing the RQs | Articles unable to address the RQs |

The inclusion criteria were chosen to ensure that only articles in English, with no temporal restrictions, addressing the relationship between Indoor Environmental Quality (IEQ) and the Sustainable Development Goals (SDGs), and capable of providing answers to the defined research questions (RQs) were considered. This ensured a more comprehensive and relevant approach to the review. The exclusion of articles in other languages was necessary to maintain consistency in understanding and analyzing the materials, while rejecting articles with missing data ensures the integrity of the information used in the review. Furthermore, excluding duplicate articles, conference papers, book chapters, conference reviews, books, editorials, and short articles focused the review on journal publications, typically peer-reviewed research sources with higher academic quality criteria, thus ensuring the relevance and reliability of the set of studies considered.

The StArt (State of the Art through Systematic Reviews) software 2.3.4.2 version, developed by the Federal University of São Carlos (UFSCAR), was used to conduct a systematic literature review following the PRISMA (Preferred Reporting Items for Systematic Reviews and Meta-Analyses) method. This software was crucial in simplifying the review process and selecting relevant articles from the databases. It facilitated the visualization of article titles, authors, and abstracts, allowing the researchers to apply the inclusion and exclusion criteria directly within the platform. In practical terms, the researchers could organize and filter articles more efficiently, quickly identifying those that met the inclusion criteria and excluding those irrelevant to the review. The StArt software saved time and simplified information management, making the review process more precise and effective. Furthermore, by following the PRISMA method, the systematic review was conducted according to internationally recognized guidelines, ensuring rigor and transparency in the selection and analysis of articles. This use of the software resulted in a more efficient quality review, as mentioned by Zamboni et al. [36].

Two different tools were used to create and visualize the bibliometric networks. The first was the VOSviewer software 1.6.17 version, which made it possible to show

the main citations, co-authorships, co-citations and co-occurrences of terminologies presented in the documents investigated [37]. The ScienceScape tool was also used as another analysis tool for developing the AKJ Sankey diagram, representing the connections between the main authors (A), keywords (K), and journals (J) [38]. In other words, it provided a visual representation capable of illustrating the flow of energy, materials, or resources applied within a system.

## 3. Results

### 3.1. Preliminary Results of the Database Search

After executing the search strategies, which included combining keywords and Boolean operators, 855 relevant articles were identified. For the selection stage and determination of their eligibility, the StArt software was used, which played a key role in managing the references and categorizing the studies according to the rigorous PRISMA method, as shown in Figure 1. This systematic approach provided a solid basis for conducting the review accurately and efficiently.

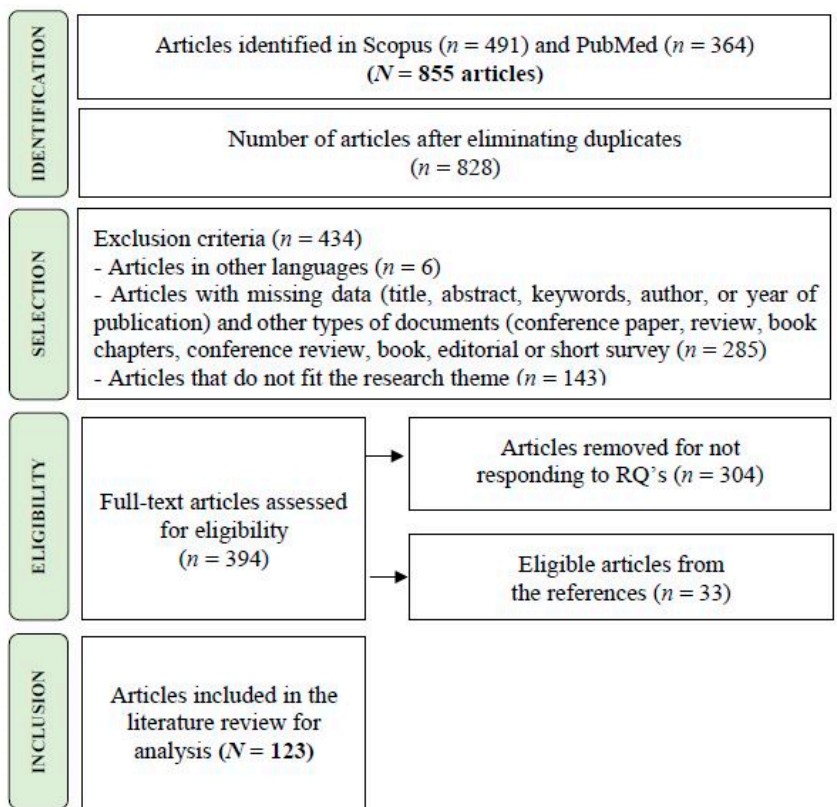

**Figure 1.** Results obtained after applying the PRISMA method.

Initially, 855 articles were identified in the Scopus database. However, after applying the exclusion criteria, 394 articles remained and were subjected to a more detailed evaluation. At this stage, each article's title, abstract and keywords were read to check their relevance to the research topic. After this analysis, 304 articles were excluded because they did not fully meet the established research questions. The result was a total of 123 articles that met the relevance criteria and were incorporated into the literature review concerning the subject in question.

### 3.2. Bibliometric Results of the Publications in the Portfolio

Through bibliometric analysis, it was possible to identify the general characteristics of the studies investigating the relationship between Indoor Environmental Quality (IEQ) and the Sustainable Development Goals (SDGs). Figure 2 displays the distribution of published

articles over the years in various journals, where each publication year is represented by stacked vertical bars composed of colored squares corresponding to the publication year. Therefore, the taller the bar for a specific year, the greater the number of publications in that year. Additionally, the vertical lines indicate each journal's respective Impact Factor (IF) for the reference year, that is, 2022. These impact factors were obtained from the Incites Journal Citation Reports of the Clarivate Analytics website [39].

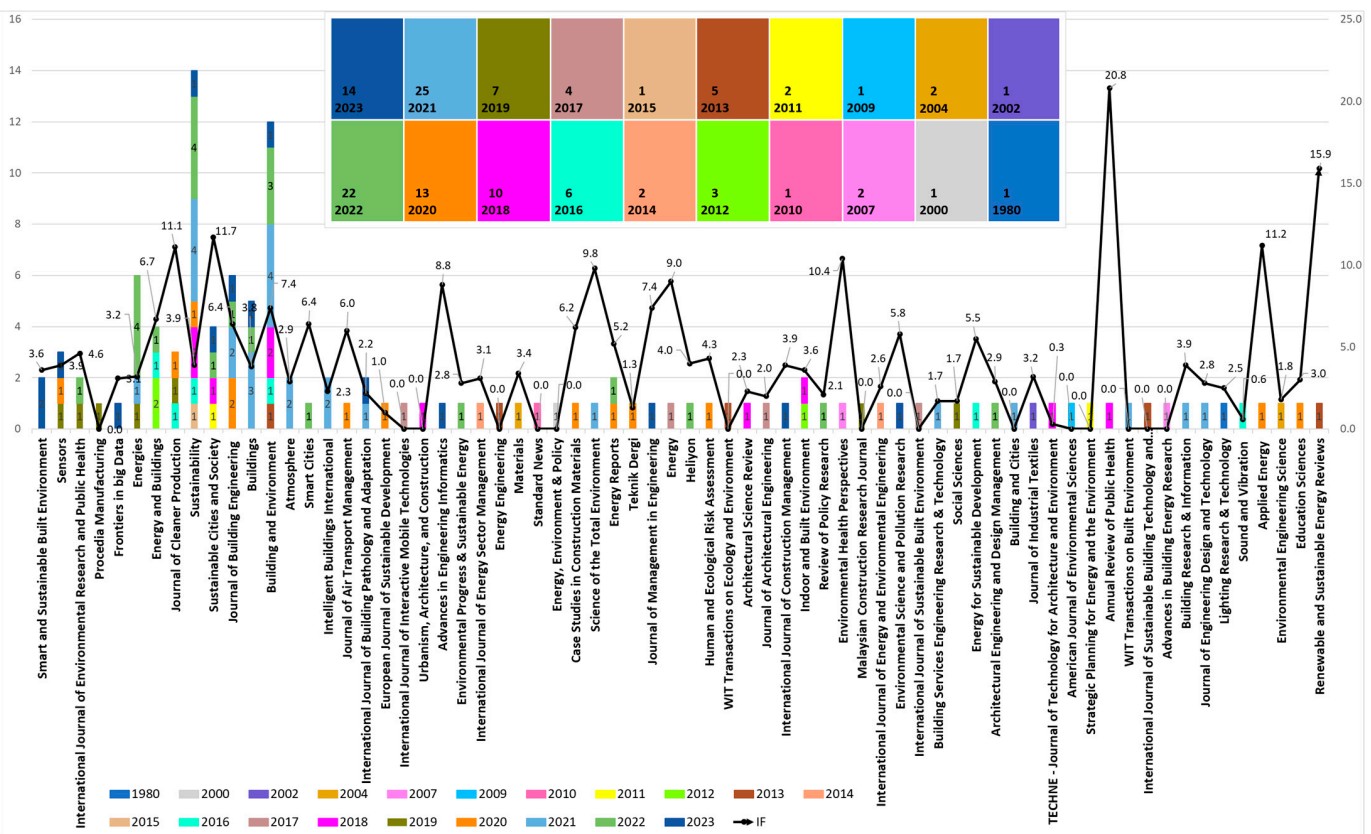

**Figure 2.** Co-occurrence map.

The analysis found that the proportion of articles was 20.33% in 2021 and 17.89% in 2022. The journals with the most publications were *Sustainability* (14) and *Building and Environment* (12), with impact factors of 3.9 and 7.4, respectively. This information provides a valuable overview of the evolution of research related to IEQ and the SDGs, as well as of the most influential journals in this area. VOSviewer software was used to create and visualize the bibliometric networks, as seen in Figure 3. The analysis revealed the presence of four clusters encompassing the most recurrent words in the articles examined. In addition, the colors and sizes of the circles highlight the importance of the words for the scope of this study.

When analyzing the clusters formed, it becomes clear that the representation of expressions is determined by the size of the circles and that the more intense colors indicate a greater frequency of occurrence. On the other hand, softer colors indicate less frequent terms.

Through a detailed bibliometric analysis, the software revealed that Cluster 3 is the most significant, highlighting the following keywords: "energy efficiency" (77), "sustainable development" (73), "buildings" (42), "quality control" (39) and "environmental management" (32). Cluster 1 is mainly made up of the keywords "indoor environmental quality" (119), "air quality" (47), "indoor air" (42), "energy utilization" (35), "environmental impact" (35) and "indoor air pollution" (30), while Cluster 2 includes terms such as "sustainability" (66), "environmental quality" (58), "building" (44), "office buildings"

(31) and "architectural design" (30). Finally, Cluster 4 is the least relevant grouping, with "environmental technology" (27) being the most prominent term. These results show that the third cluster presents the most significant connection between the topics covered in the articles, indicating an association between the topics and highlighting their relevance within the study context.

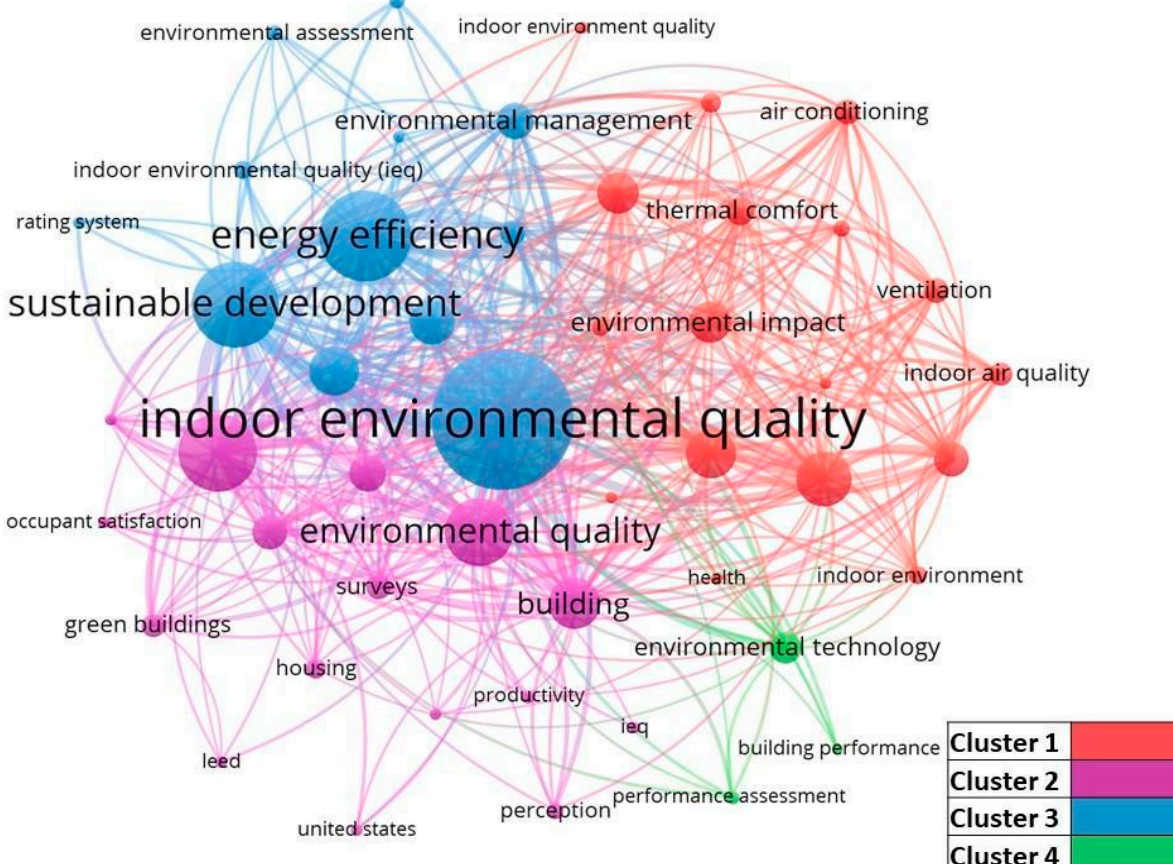

**Figure 3.** Most recurrent terms in the articles.

In Figure 4, the AKJ Sankey diagram shows the flow of information from the authors to the keywords and then from the keywords to the journals. The width of the lines in this diagram reflects the intensity of this information and its relevance within the context analyzed.

The AKJ Sankey diagram reinforces the conclusions presented using VOSviewer, highlighting the most prominent themes: "Indoor Environmental Quality", "Sustainability" and "Energy Efficiency". These themes play an essential role as the main points of connection between the authors, keywords and journals, highlighting the effectiveness of this diagram in reviewing the literature and its relationships with various areas. In addition, the diagram emphasizes the importance of sustainability in the built environment, which plays a decisive role in engineering and architecture. This connection is paramount in understanding the interactions between these themes and how they relate to developing solutions for issues concerning the built environment.

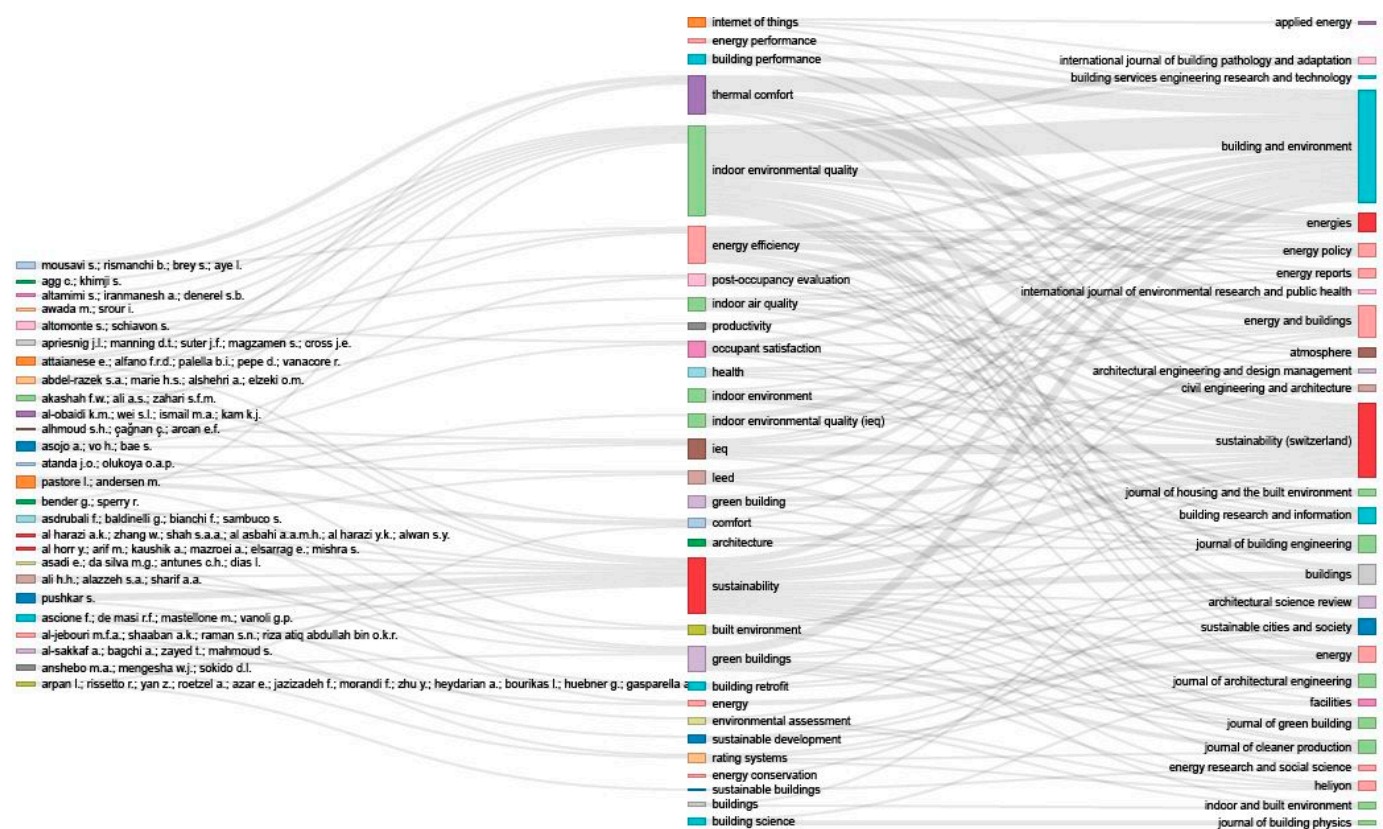

**Figure 4.** AKJ Sankey diagram.

## 4. Discussion

*4.1. RQ1: How Can Adopting Technologies and Methodologies Contribute to Efficiently Managing Indoor Environmental Quality (IEQ) in Environments, Promoting Healthy and Sustainable Conditions?*

### 4.1.1. Sustainability Assessment in Buildings

Assessing sustainability in buildings has been a topic of growing interest and debate in the construction field. As pointed out by Vilcekova, Kridlova and Burdova [5], the need to measure the impacts associated with economic, social and especially environmental performance has led to new technologies and methodologies being adopted throughout the life cycle of buildings, becoming a practice that relies on the collaboration of numerous professionals throughout the process.

A notable example of this evolution is seen in China, where Trofimova et al. [40] report on incorporating sustainable technologies into green buildings, which has become a dominant trend. Although these technologies promote energy efficiency, post-occupancy reports indicate that, in some cases, over-prioritizing this aspect can compromise occupant comfort. This leads us to reflect on the challenge presented by Asadi et al. [41], which would be to recognize which technology or methodology will be genuinely effective in the long term. Decision-makers must consider the environmental factors affecting occupant satisfaction and weigh the financial, energy, social and legal implications of adopting specific approaches.

Altomonte and Schiavon [42] establish a direct connection between this satisfaction and the IEQ, and they suggest that investigating the IEQ brings numerous benefits in terms of comfort, productivity and improved performance at work. Consequently, numerous scholars have begun to align their research with the SDGs established by the United Nations [43].

This strategic link with the SDGs reflects a comprehensive approach corresponding to the quest for human well-being and global sustainability. In this context, incorporating technologies and methodologies plays a crucial role in the effective management of IEQ,

intending to promote healthy and sustainable conditions. Among the first works found in the literature, Hunt [44] highlights the relevance of evaluating the use of artificial lighting in manually operated buildings, emphasizing the detailed analysis of the variation in illuminance levels throughout the day, showing that this type of investigation is essential for understanding the underlying patterns and optimizing the use of artificial lighting.

4.1.2. IEQ and Sustainability Integration

Brown and Gorgolewski [45] highlight post-occupancy evaluation and LEED Gold Certification, integrating occupant feedback and energy efficiency to improve IEQ and occupant satisfaction. Certifications such as Leadership in Energy and Environmental Design (LEED) and the US Green Building Council (USGBC) have been explored by authors such as Briller [46], Ibrahim [47] and Pushkar [48], offering a credit system to assess the sustainability of buildings by considering categories such as Sustainable Sites (SS), Water Efficiency (WE), Energy and Atmosphere (EA), Materials and Resources (MR), IEQ and Design Innovation. These actions aim to improve the environmental performance of buildings, promoting healthier and more ecologically responsible environments [49].

In conjunction with certifications, the sustainable building approach emphasizing natural lighting, low-toxicity building materials and adequate ventilation is discussed by Higgins, Good and Bennett [50], highlighting the importance of eco-friendly practices in creating more efficient buildings. In addition, Cedeño-Laurent et al. [51] mention the inclusion of green roofs as a strategy for promoting healthy and energy-efficient environments, aiming to improve air quality and address global challenges related to gas and pollutant emissions. Other common techniques are described by Mannan and Al-Ghamdi [52], such as the Life Cycle Assessment (LCA) and the Active Living Wall System (ALW), which explore concepts such as vertical greening and improving IAQ. This type of investment has an environmental potential capable of reducing air pollutants, providing improvements for occupants and the environment itself. In addition, the importance of raising awareness of innovative vertical greening systems and their positive impacts on indoor air and the built environment has been highlighted.

With a similar aim, Mansour and Radford [53] explore user preferences regarding design and sustainability in buildings, using Choice-Based Conjoint Analysis (CBC) to identify the relative importance of different experiential and environmental categories. This study contributes to understanding users' priorities when choosing between products and designs, reinforcing the consideration of customer preferences when developing sustainable buildings. Regarding cost reduction, Alfaris, Juaidi and Manzano-Agugliaro [54] develop an energy management program in schools, resulting in a continuous decrease in energy and greenhouse gas emissions, improving IEQ. As highlighted by Jasimin and Amat [6], green buildings offer immediate comfort to occupants and bring economic benefits and appreciation to the real-estate market. As noted by Raouf and Al-Ghamdi [55], sustainable practices conserve energy, optimize water use, promote passive heating and cooling systems, and emphasize the importance of maintenance and overhaul for the longevity of buildings.

The relationship between IEQ and sustainability is supported by Al Horr et al. [56], whose approach is the Global Sustainability Assessment System (GSAS), which can understand how buildings can be designed and operated effectively regarding human use and environmental impact. Another noteworthy aspect is the technological evolution over the years, as linked to the development of integrated photovoltaic systems (BIPV) and electrochromic glass, as explored by Bizzarri, Gillott and Belpoliti [22] and Choi et al. [57]. These technologies improve energy performance, environmental perception, the psychological health of individuals, the work experience of occupants, productivity, positive emotional responses, satisfaction with the quality of natural light, lighting conditions, and exterior views, and they reduce frequent health symptoms.

Studies such as that by Eweda et al. [14] use the Analytic Hierarchy Process (AHP) to evaluate the relationship between IEQ factors and overall building comfort by quantifying

and comparing the different elements that affect IEQ. Other significant contributions are developed by European Concerted Action (ECA) [58] and Australian Standard [59], which relate to people's dissatisfaction with the $CO_2$ concentration and noise level, highlighting the importance of air quality and acoustics in the evaluation of IEQ. In addition, it is possible to use guidelines, standards, acoustic codes and measurement of decibel levels in environments to establish acceptable noise limits more accurately, which helps to understand and effectively manage noise pollution in different spaces [60].

At the global level, Borsos et al. [61] present a comfort map that characterizes a comprehensive method, making it possible to identify the preferred workstation based on IEQ factors, promoting workers' physical and mental health. In addition, Licina et al. [62] propose IEQ Rating Systems as a comprehensive index capable of measuring IEQ through multiple criteria, integrating objective measures of the indoor physical environment and subjective perceptions of occupants, thus giving equal weight to all the components, recognizing the underlying complexity of IEQ models.

To implement these models, Gossauer and Wagner [63] highlight the importance of testing in real buildings and climate chamber laboratories, using a variety of evaluation methods, as well as applying ethnographic approaches, such as observations, photographs, notes and sketches, to improve the accuracy of field analyses [64]. Information technology is vital to assessing the built environment in field tests. Building Information Modeling (BIM) can be used from planning to operation, identifying nonconformities and problems related to quality of life and IEQ [19]. In addition, the introduction of the Smart Campus concept, which combines building information modeling with Internet of Things (IoT) wireless sensor networks, makes it possible to monitor sustainable comfort in university environments and also to share knowledge and experience to boost local sustainability projects as well as the application of Artificial Intelligence (AI) for intelligent IEQ monitoring, where the early detection of problems can be carried out, improving energy efficiency and reducing waste in buildings, which are aligned with improving living conditions and achieving SDG targets [18,65].

Thus, it is undeniable that the integration of IEQ and sustainability plays a fundamental role in the construction sector since it is one of the main consumers of energy and a significant emitter of greenhouse gases [66], and the construction industry faces the responsibility for adopting practices that balance energy efficiency and environmental impacts. In this context, innovative approaches such as the one proposed by Verma et al. [67] illustrate the search for solutions that reconcile energy consumption with occupant comfort in smart urban environments by using the Crisscross Search Particle Swarm Optimization (CSPSO) methodology and fuzzy controllers that offer a new perspective on tackling this challenge. Furthermore, CSPO can improve environmental parameters based on sensor data and user perceptions, creating a more harmonious balance between occupant needs and sustainability requirements.

Another notable integration occurs between materials and systems, as exemplified by the work of Mousavi et al. [68], which combines Phase Change Materials (PCM) and Radiant Cooled Ceilings (RCC), which are promising technologies for energy-efficient space cooling. In addition to using sensors, Yuan et al. [69] explore intelligent software simulations to predict IEQ based on Microsoft Delphi, priority weighting, and the Analytic Hierarchy Process (AHP) decision-making method. This establishes a new evaluation framework for repeatedly measuring the sustainability and health of green buildings. However, better-known simulations, such as the Computational Fluid Dynamics (CFD) employed in the research by Cheong et al. [70], make it possible to analyze IAQ and thermal comfort indoors. These simulations detail airflow and thermal conditions, identifying areas most prone to accumulating pollutants and poor air circulation, allowing ventilation and air conditioning systems to be optimized to promote occupants' health, well-being and productivity.

Another important highlight is the growing application of advanced statistical methods, such as Structural Equation Modeling (SEM), Response Surface Analysis and Bayesian Statistics, as highlighted by Nimlyat [71], Kaushik et al. [72] and Tsang et al. [73].

These methods make understanding the complex relationships between building performance and IEQ parameters possible. These quantitative approaches indicate the most influential variables in terms of the perception of visual comfort and their impact on productivity. In addition, Park et al. [74] propose a holistic approach, combining occupant perspectives, Technical Attributes of Building Systems (TABS) analysis and visual quality assessment. Complementing these statistical methods, Roskams and Haynes [75] suggest repeated sampling as a suitable approach to measuring comfort and investigating reality. This highlights the need for replicated data collection on several occasions over time to obtain more robust and reliable sampling.

Jung, Park and Ahn [4], along with Al Harazi et al. [3], highlight the importance of training and awareness in relation to sustainability. By suggesting the integration of environmental considerations into educational curricula, these studies seek to sensitize and empower university students to adopt sustainable practices. The attention paid to waste generation, reuse and recyclability contributes to training professionals who are more aware of and committed to environmental responsibility. Along the same environmental lines, Dong et al. [76] highlight the proposal for zero energy buildings (NZEB) through the relationship between energy efficiency and carbon emissions. Findings of this kind reinforce the need to adopt construction approaches that balance the energy demand and production. In this way, buildings under transient conditions are also addressed in the literature, as seen in the research by Balocco, Pierucci and De Lucia [77], in which energy performance is evaluated using an experimental method that monitors real-time experimental data related to internal and external microclimates and the flow of heat through structures. Studies on this subject offer significant insights into understanding thermal exchanges and improving energy management strategies.

Recently, the COVID-19 crisis has also influenced the energy efficiency in buildings, as discussed by Yilmaz and Yilmaz [78]. When analyzing the impact of pandemic control measures on the energy performance of buildings, an innovative probabilistic method is developed to control ventilation rates by including "modified" degree days that consider phase changes and the thermal capacity of materials. This type of analysis seeks to meet social distancing needs and minimize infection risks, highlighting the importance of adaptation in crisis scenarios and how energy efficiency strategies can be reconfigured to face unforeseen challenges.

### 4.1.3. Integration of Technologies and Methodologies for Sustainable Development in Construction

When analyzing the technologies and methodologies employed to enhance buildings, it became essential to create categories based on their purposes and applications, aiming to structure and comprehend the various approaches (Table 2).

**Table 2.** Categorization of Technologies and Methodologies Employed in Buildings.

| Category | Purpose of the Category |
| --- | --- |
| Real-time monitoring and control | Referring to the use of real-time monitoring systems to track the performance of buildings in terms of variables such as temperature, humidity, and energy consumption, among others, with the aim of optimizing their operation |
| Data analysis and simulation | Involving the use of data analysis and computer simulations to understand the behavior of buildings in different scenarios and identify improvement opportunities |
| Energy efficiency and sustainability | Encompassing technologies and practices designed to reduce energy consumption, promote cleaner energy sources, and minimize the environmental impacts of buildings |
| Comfort and environmental quality | Focusing on the creation of indoor environments that are healthy, safe, and comfortable for occupants, considering factors such as air quality, lighting, and thermal insulation |
| Post-occupancy research and evaluation | Referring to the study of building performance after occupancy, seeking to identify how design choices impact the user experience and how they can be improved |

**Table 2.** *Cont.*

| Category | Purpose of the Category |
|---|---|
| Management systems and certifications | Involving the implementation of environmental management systems and the pursuit of certifications that attest to the sustainability and energy efficiency of a building |
| Specific technologies | Focusing on specific technologies, such as smart lighting systems and efficient heating and cooling systems, among others, that contribute to the improvement of building performance |
| Diverse methodologies | Include various approaches and research methods used to analyse and improve the performance of buildings, such as case studies, life cycle analysis, etc. |
| Awareness and education | Including various approaches and research methods used to analyze and enhance building performance, such as case studies and life-cycle analysis, among others |

The division of these categories provides a comprehensive overview of the various approaches to improving buildings, highlighting the technologies and methodologies that contribute to energy efficiency, sustainability, quality of life and general awareness of the importance of responsible and innovative construction through the lens of sustainability. In addition to categorizing them, the studies that answered this research question were also connected to the Sustainable Development Goals (SDGs), as illustrated in Figure 5.

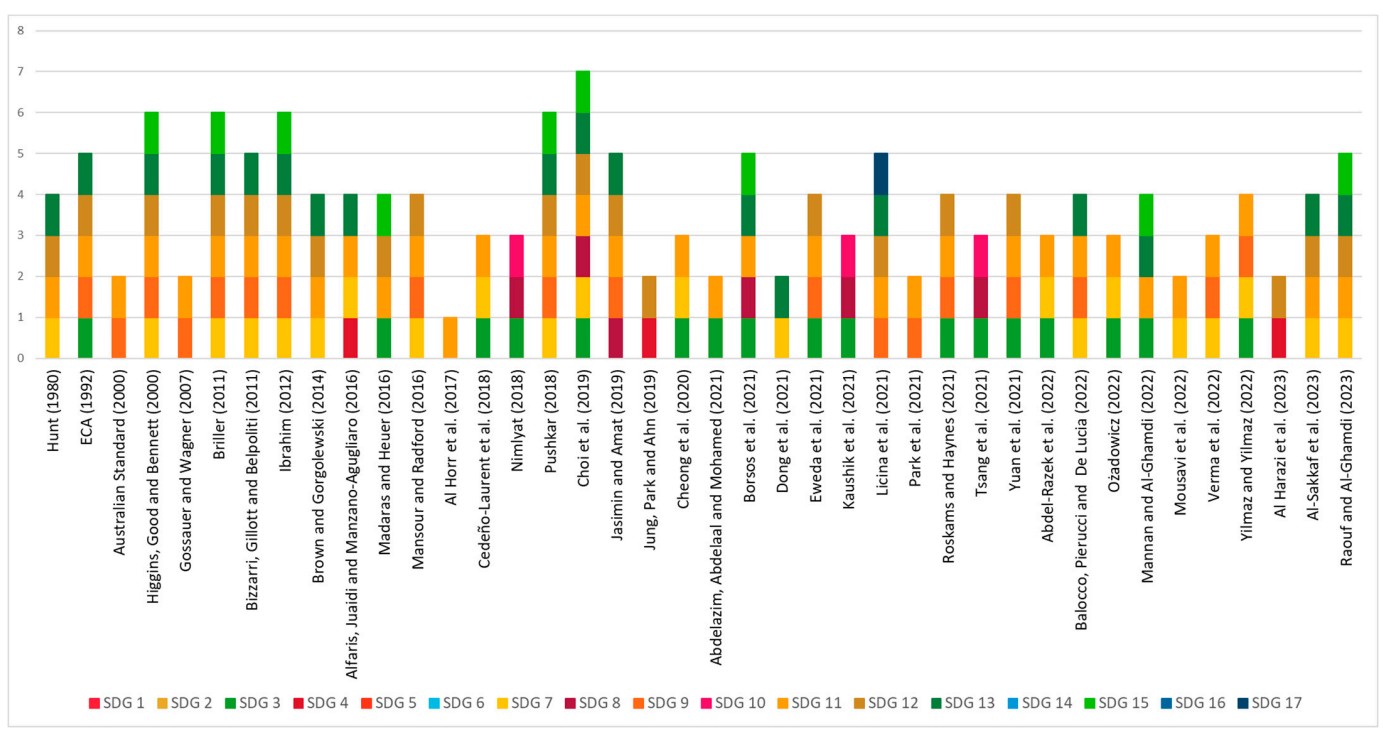

**Figure 5.** Linking the studies to the Sustainable Development Goals (SDGs) [3,4,6,14,18,19,22,44–48,50–63,65–78].

When examining the studies that could address this research question, it was observed that out of the 40 articles discussing technologies and methodologies, it was possible to identify a proportion of research related to specific Sustainable Development Goals (SDGs) as follows: SDG 11 (85%), SDG 7 (52.50%), SDG 12 (50%), SDG 9 (45%), SDG 13 (45%), SDG 3 (42.5%), SDG 15 (22.50%), SDG 8 (15%), SDG 4 (7.50%), SDG 10 (7.50%), and SDG 17 (2.50%). After analyzing the studies, it became clear that technologies and methodologies are predominantly linked to the goals of SDG 11—Sustainable Cities and Communities (85%), SDG 7—Affordable and Clean Energy (52.50%), SDG 12—Responsible Consumption and Production (50%) and SDG 9—Industry, Innovation and Infrastructure (45%).

This assessment identified how priorities focus on the development of more sustainable and resource-efficient urban planning, the creation of environments designed for

people's well-being, following quality and comfort standards, the construction and operation of buildings with an emphasis on energy efficiency and sustainability to reduce the consumption of natural resources and carbon emissions, the management of construction waste, the use of certifications and standards, energy monitoring and control, awareness-raising and education, the adoption of advanced technologies and the development of smart cities.

Among the studies that simultaneously addressed the SDGs, we highlight the research by Briller [46], Ibrahim [47] and Pushkar [48], who used certifications such as Leadership in Energy and Environmental Design (LEED) and the US Green Building Council (USGBC) to improve the environmental performance of buildings, promoting healthier and more ecologically responsible environments. In addition, such research has also been linked to SDGs 13 and 15, with similar objectives concerning energy efficiency, renewable sources, sustainable urban planning and low-carbon building materials to reduce greenhouse gas emissions and environmental education.

The connection between IEQ and sustainability in the construction industry has been emphasized throughout this section. This approach seeks to promote human well-being and global sustainability, exploring strategies ranging from IEQ management and adopting certifications and sustainable practices in construction to implementing technologies such as BIM and IoT for monitoring and applying advanced statistical methods. Awareness is highlighted as a crucial factor, along with adapting energy efficiency strategies to address the challenges posed by COVID-19. Integrating IEQ and sustainability is central to promoting healthy environments and energy efficiency, which is in line with the Sustainable Development Goals.

*4.2. RQ2: How Can the Assessment of Environmental Impacts and Risks Be Applied to Guarantee Indoor Environmental Quality (IEQ) in Public and Private Environments When following the Principles of Sustainable Development?*

4.2.1. Evolution of Building Priorities and Strategies over Time

Assessing environmental impacts and risks in the construction sector directly relates to the SDG targets. This link is motivated by the construction industry being one of the largest energy consumers and a significant source of pollution. These problems have been exacerbated over time due to climate change, which has increased the demands on the sector and compromised the integrity of built structures [79]. According to Licina et al. [55], assessment priorities in buildings have changed over time, especially in the green building industry, as shown in Figure 6.

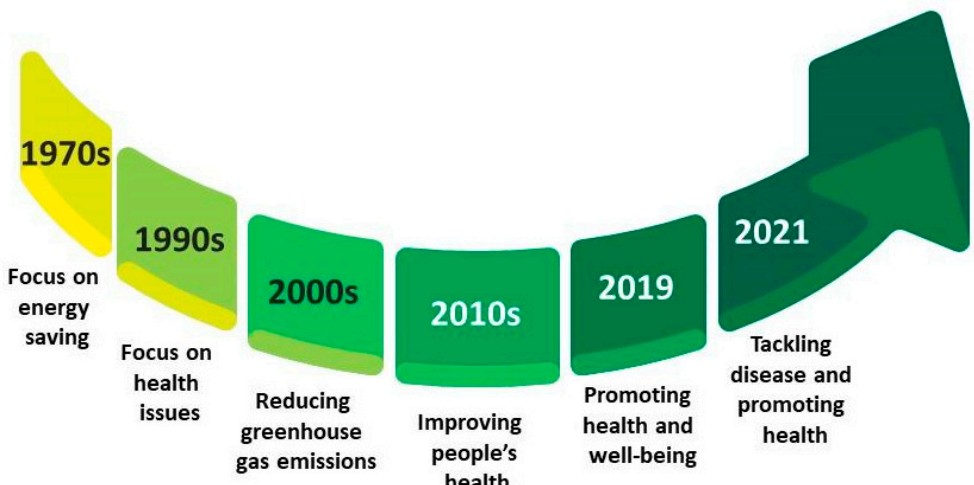

**Figure 6.** Timeline of green building priorities.

The evolution of building priorities over time reflects changes in social, environmental, technological and scientific perspectives. Each decade, different emphases have been placed

on evaluations and objectives to make buildings more sustainable and healthier. In the 1970s, Rey et al. [80] relate the focus on energy saving to the oil crisis, which caused a significant increase in energy costs, generating the need to reduce consumption in buildings and fostering the search for ecologically friendly solutions. These concerns about energy efficiency began to influence the design and construction of buildings, focusing on thermal insulation, controlled ventilation and more efficient heating and cooling systems [81].

During the 1990s, attention turned to health issues in relation to buildings. Indoor air quality and other factors impacting the health of occupants gained prominence. Sick Building Syndrome (SBS) emerged as an important concept, considering air quality, low-emission building materials, humidity control and thermal comfort. Buildings have been designed to minimize exposure to chemical substances and allergens to improve users' well-being [82–84].

In the early 2000s, increased concerns about climate change, including global warming, led to the priority of reducing greenhouse gas emissions from buildings. Efforts were focused on developing low-carbon facilities, incorporating technologies such as solar panels, energy recovery systems and sustainable construction strategies [85,86] to achieve energy savings, pollutant control and better performance [82].

Since the 2010s, the focus has been on human well-being and occupants' quality of life. Projects began to consider elements such as natural lighting, healthy indoor environments, adequate acoustics and connections with nature. Certifications such as LEED started to address not only environmental sustainability but also human comfort [87,88], as did the Oman Building Environmental Certification (OMBEC) in Oman, which encompasses IEQ, water issues, waste management, and integrated development [89]. Customized simulation strategies have also contributed to this theme, such as the adaptive insulation simulation model, which improves occupant comfort [90] as well as architectural projects in Egypt that consider the surrounding environment and air quality by using reinforced concrete and Karshif particles [91] and the use of lignocellulosic bio-waste as sustainable thermal insulation alternatives [92].

In 2019, there was a shift in focus, expanding beyond physical well-being to encompass health's psychological and social aspects. Design strategies were implemented to foster social interactions, reduce stress and increase productivity. The aim was to create environments that supported occupants' mental and emotional health. Among these approaches, in Canada, Soudian and Berardi [93] developed a multifunctional façade for buildings capable of regulating the flow of heat, humidity and air. On the other hand, in Greece, Mastellone et al. [94] highlighted the implementation of energy and structural retrofitting, resulting in significant improvements in IEQ, especially in terms of thermal comfort and air quality.

The situation became more evident in 2021 due to the COVID-19 pandemic, which began in 2020 [95]. Consequently, there has been a shift in focus toward disease prevention strategies. This has led to the development of ventilation systems, antimicrobial surfaces, layouts that promote physical distancing and the integration of health technologies into buildings. These measures were taken in response to the need to create safer and healthier environments in the face of the pandemic.

### 4.2.2. Environmental Impact and Risk Assessment for Sustainable Building Performance

Research and knowledge development and electronic automation are emerging as fundamental technologies and techniques for planning, operating, and constructing sustainable buildings. In this context, Broday and Gameiro da Silva [96] emphasize that Industry 4.0 has played a pivotal role in reshaping the built environment, mainly through the introduction of robotics and digitization to enhance building health. However, studies indicate a lack of research and profound understanding of the qualitative connections between Indoor Air Quality (IAQ), workplace performance or health, and their relationship with energy consumption and energy efficiency. To address this gap, research focuses on modeling the performance of healthy and sustainable buildings. This encompasses the exploration of



assessment methods, metric analysis, and the identification of essential interconnections. Furthermore, synergistic sustainability criteria and indicators are being identified [97].

Attaianese et al. [28] point out that as this issue is recent in the scientific community, unfortunately, approaches tend to be isolated. As research into IEQ has advanced, experts have realized the need to develop comprehensive certifications, standards and guidelines. These initiatives not only address technical issues but also consider the contractual, legal and procedural dimensions throughout the entire life cycle, including design, operation [52,98], remodeling [99] and refurbishment [100]. These approaches have been shown to increase occupant satisfaction [21].

Another relevant aspect is addressed in Ascione et al.'s research [101], which draws attention to the lack of consideration in many certifications regarding crucial aspects of the built environment, such as seismic risks, environmental threats, climate change, urban heat islands, and external light and noise pollution. This underscores the urgency of reviewing or establishing new standards, especially concerning ventilation, which has gained prominence due to the COVID-19 pandemic [102]. Lee and Park [103] emphasize that biophilic design has been the subject of increasing research due to the pandemic, climate change, and environmental concerns, aiming to enhance the quality of life in environments by providing users with better experiences and more positive emotions.

Tham [104] points out that, among various concerns, Indoor Air Quality (IAQ) is of the utmost importance due to the presence of contaminants from both internal and external sources, such as polluting gases, particles, furniture, air conditioning systems, and humidifiers. These elements can be harmful to health, especially in densely populated environments and when synthetic materials are used. Adopting air purification and circulation techniques, smart building technologies, and reducing polluting activities have contributed to improving quality of life, well-being, productivity, and satisfaction [82,100,105–108].

Air pollution is undoubtedly a significant threat to human health and global warming. This is due to various atmospheric pollutants, including carbon monoxide, $PM_{10}$, $PM_{2.5}$, nitrogen dioxide, ammonia, and ozone. A notable example is sulfur dioxide, which triggers severe respiratory diseases associated with burning fuels and seriously impacts agriculture. In the industry and transportation sectors, nitrogen dioxide and carbon monoxide are common problems, harming both human respiratory systems and vegetation. Carbon monoxide, for example, is linked to headaches and respiratory problems, primarily due to waste burning and transportation use. Another critical pollutant is ozone, which forms from photochemical processes and can cause premature aging. Additionally, $PM_{10}$ and $PM_{2.5}$ are particles that pose significant health risks to individuals [109].

A noteworthy study conducted by Hernandez et al. [110] in Quito, Ecuador, emphasizes the importance of investigating Indoor Air Quality (IAQ) through the measurement of the particulate matter ($PM_{2.5}$) concentration in La Carolina Urban Park. This park, located in a residential, commercial, and entertainment area, demonstrated the ability to act as a natural filter for atmospheric pollutants. This underscores the relevance of preserving green spaces in cities, improving transportation systems to reduce pollutant emissions, and implementing more effective vehicle traffic management systems. These points highlight the urgent need to address air pollution on multiple fronts, from government regulation to individual actions, to protect both people's health and our planet.

In energy terms, waste management and energy consumption are very prominent in the construction process [111], and they have objectives related to creating ideal IEQ conditions that can co-occur with the active participation of occupants in improving site conditions [112]. Among the evaluation alternatives, benchmarks can be used to investigate water capacity, energy capacity and even carbon intensity [2,113]. By analyzing these conditions, it becomes possible to explore the ideal design of sustainable environments [114] and better use of resources [115].

The search for more sustainable built environments has become a priority worldwide, driven by the desire to guarantee people's quality of life and preserve natural resources for future generations. To achieve these goals, Environmental Impact and Risk Assess-

ment has emerged as an essential tool, making it possible to assess and mitigate the adverse effects of human activities on public and private environments via the principles of sustainable development.

The application of Environmental Impact and Risk Assessment is evident in a study that identifies sustainability indicators to evaluate green buildings in Malaysia using the Green Building Index (GBI). In this context, "Energy Efficiency" and "Indoor Environmental Quality" emerge as critical indicators, reflecting the importance of considering the external environmental impact and the occupants' experience inside buildings. The emphasis on these indicators reinforces the need to minimize energy consumption and provide healthy and comfortable indoor environments in line with the principles of sustainable development [116].

In Romania, the emphasis on energy efficiency has led to a broader focus on improving indoor air quality, especially regarding home radon mitigation. This study exemplifies how Environmental Risk and Impact Assessment can be employed to identify specific risks and implement solutions that address both energy efficiency and occupant health issues. Combining traditional approaches with modern indoor air monitoring technologies demonstrates a commitment to long-term sustainability [117].

As demonstrated in a study carried out in Ethiopia, creating a comprehensive assessment tool to measure the sustainability of buildings emphasizes the need to consider not only environmental aspects but also economic and social ones. This research highlights the complexity of sustainable issues in construction, especially in contexts of limited resources. Environmental Impact and Risk Assessment emerges as an approach that can integrate multiple dimensions of sustainability, allowing for more holistic and informed assessments [118,119].

The South Korean scenario, where school buildings face indoor air quality and thermal comfort challenges, highlights how Environmental Impact and Risk Assessment can be employed to improve educational environments. The HVAC system control strategy proposed in this study demonstrates the practical application of risk assessment to enhance environmental quality and energy efficiency. This reinforces the importance of creating healthy indoor spaces to improve the well-being of occupants [120].

A study in Spain, which focuses on cooperative housing, offers insights into how Environmental Impact and Risk Assessment can be adapted to specific housing contexts. The analysis of energy impact and resident comfort highlights how sustainability can be approached in an integrated way, considering factors such as energy consumption, occupant satisfaction and IEQ. This highlights the relevance of Environmental Impact and Risk Assessment in evaluating the sustainable performance of different types of buildings and promoting continuous improvements [121].

Research into rural development in China highlights a gap in the conventional approach to sustainability assessment, which often focuses predominantly on the environmental performance of buildings. The inclusion of the social dimension becomes crucial, especially in rural environments. This discussion highlights the need to adapt Environmental Impact and Risk Assessment to address specific challenges, ensuring that local communities benefit from built environments that are sustainable not only in environmental terms but also in social and economic terms [122].

In this sense, Montiel et al. [123] argue that there is an intrinsic link between the underlying purposes and the structures of buildings. This is established by associating energy efficiency, thermal comfort, visual quality and air purity in interior spaces. In addition, Calvo et al. [27] highlight the importance of incorporating monitoring systems that identify uncomfortable conditions within these spaces. This way, a path is outlined for future interventions to optimize the systems, resulting in full compliance with the SDGs. From a similar perspective, Korsavi, Jones and Fuertes [124] mention the crucial building of energy awareness from the earliest years of life, particularly in educational environments. In these scenarios, it is possible to include from an early age the presence of adaptive behaviors that

contribute to improving IEQ, generating a cascading effect that increases overall comfort and substantially reduces energy consumption.

Environmental Impact and Risk Assessment has emerged as an essential tool for guaranteeing indoor environmental quality in public and private environments when following the principles of sustainable development. The studies discussed demonstrate how this approach can be applied holistically, considering sustainability indicators, mitigating specific risks, promoting improvements in the quality of the indoor environment and integrating social, economic and environmental dimensions, contributing to building a more sustainable and resilient future.

*4.3. RQ3: What Are the Main Challenges and Opportunities in Environmental Management and Policy to Promote Indoor Environmental Quality (IEQ) in Urban Environments, Considering the Balance between Sustainable Development and Environmental Preservation?*

According to Gallo and Romano [125], urban areas consume around 75% of all global energy resources, resulting in approximately 80% of polluting gas emissions, thus representing one of the main challenges facing communities. These challenges include financial constraints and citizens' expectations, as well as the need to attract investment and create job opportunities to make these locations more sustainable, attractive, efficient and livable [2]. Along the same lines, Niza et al. [126] point out that the construction sector plays a significant role in emitting pollutants due to a lack of planning, excessive energy consumption for heating and waste of resources. Faced with this scenario, governments have recommended the establishment of environmental targets aimed at reducing the emission of pollutants and have also encouraged the adoption of more sustainable behavior by individuals.

In this context, governments have recommended the establishment of environmental goals aimed at reducing pollutant emissions and have encouraged the adoption of more sustainable behaviors by the population. This aims to establish a new paradigm in construction and building practices [127], which can be implemented following regulations set by the World Health Organization (WHO) and the Environmental Protection Agency. However, challenges persist in monitoring and controlling air quality through more cost-effective approaches [128]. It is essential to highlight that Indoor Air Quality (IAQ), as emphasized by Gola, Settimo, and Capolongo [129], is one of the main concerns of governments because it is directly related to people's health. Improving IAQ is one of the essential goals of sustainable development. In addition, there is concern about air pollution since it plays a significant role in the spread of airborne diseases. In this sense, the renovation of buildings can optimise energy efficiency and the well-being of individuals [130].

Another significant issue that requires attention from authorities is noise pollution. This matter negatively impacts the quality of life in urban areas, the health of citizens and property values. Therefore, political leaders are increasingly committed to reducing the noise impact to make cities more sustainable and pleasant [131]. In this context, Vladimir and Madalina [132] propose that the best long-term approach is the implementation of effective urban planning, with an emphasis on sustainable design to control mechanical noise in projects, especially in sectors such as the hotel industry, which relies on high-energy-consuming HVAC systems that are prone to noise propagation [133].

Furthermore, it is important to note that the issue of noise pollution is not limited to urban areas but also affects airports, which significantly contribute to greenhouse gas emissions, have impacts on the soil and passengers' health, and generate noise pollution. Thus, adherence to the Sustainable Development Goals (SDGs) provides an opportunity to reconcile economic development with environmental preservation. However, achieving this balance requires effective collaboration between governments, civil society organizations, and private sector companies [26].

In this context, Jung, Park and Ahn [4] highlight the need to raise awareness among individuals of atmospheric changes, climate issues and global warming, promoting efforts to ensure greater preparedness in the face of challenges and stimulating the development of innovative ideas [134]. Thus, the proposal to offer training emerges as an opportunity

for them to promote their well-being through more in-depth education and communication related to construction and design practices [135]. This is exemplified by the offer of the "Indoor Environmental Comfort in Buildings (IECB)" course in the distance learning modality, which is based on the Indoor Environmental Quality (IEQ) discipline of the master's and doctoral program in Energy for Sustainability in Portugal [136].

Globally, the green building trend has gained prominence for addressing these environmental issues. However, it is essential to emphasize the need to recognize the importance of these buildings for IEQ. In addition, in developing countries in particular, green building codes do not yet cover this issue, showing a limitation. Another noteworthy point is that transitioning from a conventional building to a green building will not always result in performance improvements, which represents a challenge for increasing occupant satisfaction [137]. Therefore, developers, designers and policymakers must focus intensely on occupant-related aspects to improve health and comfort. In this sense, implementing a continuous management plan can be a viable alternative, as well as renovating and revitalizing historic buildings to integrate sustainability principles, considering the preservation of both their historical and cultural value [138].

Ahn et al. [16] note that applying technologies, advanced methodologies, tools and high-performance workstations faces significant challenges, especially related to financial costs. In many places, these approaches are unfeasible due to the primary conditions in which people live in situations of vulnerability, making facilities such as air-conditioning systems inaccessible, for example [10,139,140]. Furthermore, implementing strategies aligned with the SDGs is not trivial, as Luerssen et al. highlight [141]. They argue that economic growth is not achievable for every population, especially in remote or rural areas, which results in a lack of resources affecting areas such as health and well-being (SDG 3), availability of water resources and sanitation (SDG 6), clean and affordable energy (SDG 7), and consequently, the need to mitigate inequalities (SDG 10).

According to Jiang and Kurnitski [142], despite global interest in sustainable development, the issue is still not adequately monitored within universities due to difficulties using sustainable design tools in these environments. In this way, Olsson, Malmqvist and Glaumann [143], through interviews, find that there are obstacles to the sustainable renovation of buildings, mainly due to the lack of sustainability guidelines and scarce knowledge on the subject. In addition, the authors highlight the need for government support, incentives, and the adoption of new business models that prioritize environmental factors.

In Mexico, for example, Saldanã-Márquez et al. [144] present the development of the Financing Program for Housing Solutions (FPHS), which aims to build sustainable social housing capable of integrating features related to IEQ, energy efficiency and management. In this context, the public sector assumes responsibility for incorporating aspects of the urban environment and providing a plan enabling low-income families to acquire suitable housing. From a similar perspective, Al Harazi et al. [3] discuss the situation in Yemen, where the construction industry is still exploring cost-effective strategies to meet the population's housing needs, often still resorting to traditional methods and neglecting sustainability. In Nigeria, Atanda and Olukoya [29] emphasize the urgency of implementing measures to mitigate the environmental impacts caused by buildings.

In Algeria, Djebbar, Salem and Mokhtari [145] present thermal rehabilitation procedures to be carried out in different timeframes for residential areas. On the other hand, the situation in the United States is different, as governments strive to link housing, sustainability and community development. Through these initiatives, it is possible to analyze the challenges and opportunities of addressing these interdisciplinary issues [84]. The European scenario illustrates progress in sustainable design strategies by considering the life cycle of neighborhoods and communities and the growing importance of architectural approaches centered on human needs, especially emphasized by the COVID-19 pandemic [146]. Therefore, the quest for sustainability in the building sector must continue

to evolve, embracing the complexity of interconnected issues and promoting coordinated actions that transcend disciplinary and geographical boundaries.

*4.4. Future Trends and Research Gaps*

In general, companies increasingly need to develop strategies that improve efficiency, satisfaction and health, aiming for sustainable returns in the long term [147]. In this sense, the literature has identified future trends, including integrating the IoT, sensors and High Dynamic Range (HDR) imaging techniques to monitor IEQ parameters [27,102,148] to increase the measurement accuracy. In addition, there is the development of real-time ventilation control algorithms [140], personalized ventilation innovations, sensing and management [104], and the computational analysis of air distribution in the environment employing Computational Fluid Dynamics (CFD) [149] to optimize the ventilation rate and air quality [150,151] and thus reduce the incidence of allergic and infectious diseases and Sick Building Syndrome (SBS), among others [82].

Regarding sustainability, energy renovation stands out for its thermal and visual improvements in various buildings and climatic conditions [152]. In addition, integrating renewable energy sources and energy efficiency technologies is crucial to achieving energy neutrality and reducing carbon emissions [76]. Innovative systems are also emerging that bring together new technologies and intelligent materials in the construction and renovation of buildings [125], as well as the adoption of environmentally sustainable interior design practices [153], including the implementation of green buildings, ecologically responsible technologies and sustainable construction guidelines to improve performance [56,62,154].

Another trend is the growing focus on occupant health, which leads to an increase in wellness-oriented building design [135], along with investments in Post-Occupancy Evaluation (POE) [155]. The study of the experience of professionals is conducted to explore IEQ and quality of life at work, especially concerning social interactions [156]. There is also an increase in the use of eco-efficient materials and products derived from recyclable resources and rapidly renewable sources with a low environmental impact [55]. A holistic approach that integrates health, social, environmental and construction sciences plays a crucial role [62], as does implementing training programs to raise awareness of sustainable construction practices and design, enabling better selection of resources, materials and sites, promoting social responsibility and the adoption of ecologically conscious approaches [4]. Furthermore, scientific research into the impacts of IEQ on health and well-being becomes relevant to identify risks, highlight benefits and explore other aspects [84].

**5. Conclusions**

This research focuses on technological advances and challenges in managing healthy and sustainable environments, focusing on the interaction between Indoor Environmental Quality (IEQ) and the Sustainable Development Goals (SDGs). A significant research effort involves assessing human responsibility for climate change and waste generation and developing sustainability to preserve the planet and human needs. A more comprehensive approach to the parameters of the built environment would make it easier to identify solutions to improve site sustainability and achieve the SDGs more effectively. When exploring the Scopus and PubMed databases, it was decided to avoid imposing any restrictions to cover the literature more thoroughly. As a result, we observed the highest concentration of publications in 2021, corresponding to 20.33% of the documents in the portfolio analyzed, totaling 25 articles. This article proposed three research questions (RQs). In response to RQ1, several results were found, including:

- Adopting technologies and methodologies is crucial for improving the management of Indoor Environmental Quality (IEQ) in spaces, promoting the creation of healthy and sustainable environments.
- New technologies are vital in assessing economic, social, and environmental impacts.
- The Global Sustainability Assessment System (GSAS) and similar tools are essential for the effective design and operation of buildings.

- The potential of technologies in relation to the energy performance of buildings and the health of occupants.
- The development of these technologies and methodologies is aligned with the Sustainable Development Goals (SDGs).
- The connection between Indoor Environmental Quality (IEQ) and sustainability is important in creating healthy environments and promoting energy efficiency.
- Strategies and challenges associated with this theme, emphasizing considerations related to the COVID-19 pandemic.

In line with RQ1, the responses to RQ2 identified the following:

- Evaluating environmental impacts and risks is crucial to ensuring Indoor Environmental Quality (IEQ) in public and private environments, which is in line with sustainable development principles.
- Over the decades, there has been an evolution in environmental impact assessment, encompassing everything from energy savings to human well-being and mental health.

- Due to the pandemic, there has been a shift in strategies for disease prevention in the built environment.
- Environmental Impact and Risk Assessment is crucial in shaping a sustainable and resilient future.

In response to RQ3, it was identified that the main challenges in environmental management and policy to promote Indoor Environmental Quality (IEQ) in urban environments include:

- Challenges related to excessive resource consumption and environmental emissions.
- The need to make urban areas more sustainable and efficient.
- Adopting environmental goals and sustainable behaviors, thereby aligning with the Sustainable Development Goals (SDGs), provides opportunities to balance economic development and environmental preservation.

Several promising research areas were identified to address the gaps identified based on the obtained results. These include investigating the long-term impacts of the adopted technologies and methodologies for improving Indoor Environmental Quality (IEQ) on sustainability goals and human well-being, exploring the development of innovative technologies and methodologies specifically designed to address the challenges of managing healthy and sustainable indoor environments, conducting a more in-depth analysis of the effectiveness and applicability of global sustainability assessment systems, examining the pandemic's implications for strategies to ensure IEQ in buildings by investigating long-term changes in building design, occupancy patterns, and health-related practices, exploring the role of human behavior and occupant engagement in creating sustainable indoor environments, analyzing the integration of IEQ considerations into urban development and governance, as well as assessing the cost-effectiveness of different technologies and methodologies, considering both short-term investments and long-term savings. Furthermore, encouraging interdisciplinary research collaborations among environmental scientists, architects, engineers, public health experts, and policymakers is crucial to comprehensively addressing the complex challenges related to IEQ and sustainable development.

This research has delved into the technological advances and challenges of managing healthy and sustainable environments, focusing on the connection between IEQ and the SDGs. At a time when global awareness of climate change and waste generation is at an all-time high, this research becomes even more relevant by investigating ways of reconciling the planet's preservation with humanity's needs.

**Author Contributions:** Conceptualization, I.L.N., A.M.B. and E.E.B.; methodology, I.L.N.; software, I.L.N.; validation, I.L.N., A.M.B. and E.E.B.; formal analysis, E.E.B.; investigation, I.L.N. and A.M.B.; resources, I.L.N. and A.M.B.; writing—original draft preparation, I.L.N. and A.M.B.; writing—review

and editing, I.L.N., A.M.B. and E.E.B.; visualization, I.L.N., A.M.B. and E.E.B.; supervision, E.E.B.; project administration, E.E.B.; funding acquisition, I.L.N. All authors have read and agreed to the published version of the manuscript.

**Funding:** This research was funded by the "Coordenação de Aperfeiçoamento de Pessoal de Nível Superior" (CAPES)—financing code 001 and "Conselho Nacional de Desenvolvimento Científico e Tecnológico" (CNPq).

**Data Availability Statement:** Not applicable.

**Conflicts of Interest:** The authors declare no conflict of interest.

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
