# Peer review of "Indoor Environmental Quality (IEQ) and Sustainable Development Goals (SDGs): Technological Advances, Impacts and Challenges in the Management of Healthy and Sustainable Environments"

_urbansci, doi:10.3390/urbansci7030096_

Round 1

Reviewer 1 Report

Comments for authors

·         Title. Please remove the capital letter of Technological advances.

·         Abstract. Try to include the research questions for which conclusions are presented.

·         Introduction. IEQ is not well defined, and it is introduced only in line 59. Please add a clear definition of IEQ that comprehends the main category used for the evaluation, the POE protocol and lasty the direct or indirect weighting process of the category themselves that changes substantially results of the evaluation. For this distinction you can refer to:

§  Lolli, F., Coruzzolo, A.M., Balugani, E.: The Indoor Environmental Quality: A TOPSIS-based approach with indirect elicitation of criteria weights. Saf. Sci. 148, 105652 (2022). https://doi.org/10.1016/j.ssci.2021.105652.

 In addition, relevant reviews on the theme should be cited in the introduction to clarify the novelty of the research related to current literature and to point out the main characteristics of an IEQ evaluation. The following papers, but not only these ones, should be cited for the scope:

o   D. Kolokotsa , M. Santamouris, : Review of the indoor environmental quality and energy consumption studies for low income households in Europe,D. Kolokotsa , M. Santamouris, https://doi.org/10.1016/j.scitotenv.2015.07.073.

o   Lolli, F., Marinello, S., Coruzzolo, A.M., Butturi, M.A.: Post-Occupancy Evaluations (POE) Applications for Improving Indoor Environment Quality (IEQ). Toxics 2022, Vol. 10, Page 626. 10, 626 (2022). https://doi.org/10.3390/TOXICS10100626.

o   M. Ortiz, L. Itard, P.M. Bluyssen: Indoor environmental quality related risk factors with energy-efficient retrofitting of housing: a literature review. Energy Build., 221 (2020), Article 110102

·         Materials and Methods. The review is based only on Scopus, please check if other references are available on Google Scholar or PubMed. In addition, please clarify here on in the introduction the meaning of “adopting technologies”. In general, it seems like most of concepts here are given for granted but a clear specification is needed.

·         Results. Please check if the number of articles excluded for missing data is correct since it’s more than half of collected paper and seems strange. Figure 3: please write the full name of journals for publication in respect to MDPI standards. Figure 5: please try to upload a higher quality image since without zooming is very difficult to read. Figure 6: I think that the introduction about SDGs should be pointed out in the introduction Section not here in the results. Here you also introduced AHP, please in the introduction cite it as one of the direct methods to elicit category weights. Please make a more detailed categorization of technologies applied to buildings starting from Figure 7. In particular, at least try to link specific technology to one or more of the individuated categories. Future trends and research gaps: please try to focus more on the gaps since as it is this section seems to be a collection of new technologies applied for IEQ evaluation.

·         Conclusion: try to summarize the findings of the three research questions into a bullet list. In addition, future research to fill the individuated gaps should be proposed since is one of the scopes of a literature review to guide researchers in their future works.

English is quite good minor editing is required. 

Author Response

Reviewer 1:

Dear reviewer, thank you very much for your time and your comments in order to improve this paper. Please see below the answers to all your comments.

Comments:

1) Title. Please remove the capital letter of Technological advances.

The capital letter in “technological advances” has been removed as requested.

2) Abstract. Try to include the research questions for which conclusions are presented.

Incorporating the three research questions into the abstract becomes unfeasible due to its length, given that the abstract already contains 215 words.

3) Introduction. IEQ is not well defined, and it is introduced only in line 59. Please add a clear definition of IEQ that comprehends the main category used for the evaluation, the POE protocol and lasty the direct or indirect weighting process of the category themselves that changes substantially results of the evaluation. For this distinction you can refer to:

  • Lolli, F., Coruzzolo, A.M., Balugani, E.: The Indoor Environmental Quality: A TOPSIS-based approach with indirect elicitation of criteria weights. Saf. Sci. 148, 105652 (2022). https://doi.org/10.1016/j.ssci.2021.105652.

In addition, relevant reviews on the theme should be cited in the introduction to clarify the novelty of the research related to current literature and to point out the main characteristics of an IEQ evaluation. The following papers, but not only these ones, should be cited for the scope:

  • Kolokotsa, M. Santamouris,: Review of the indoor environmental quality and energy consumption studies for low income households in Europe,D. Kolokotsa , M. Santamouris, https://doi.org/10.1016/j.scitotenv.2015.07.073.
  • Lolli, F., Marinello, S., Coruzzolo, A.M., Butturi, M.A.: Post-Occupancy Evaluations (POE) Applications for Improving Indoor Environment Quality (IEQ). Toxics 2022, Vol. 10, Page 626. 10, 626 (2022). https://doi.org/10.3390/TOXICS10100626.
  • Ortiz, L. Itard, P.M. Bluyssen: Indoor environmental quality related risk factors with energy-efficient retrofitting of housing: a literature review. Energy Build., 221 (2020), Article 110102

All the suggested articles have been cited throughout the introduction, adding a clear definition of IEQ, the POE protocol, weighting processes, energy poverty, and other aspects to fulfill all the requests.

4) Materials and Methods. The review is based only on Scopus, please check if other references are available on Google Scholar or PubMed. In addition, please clarify here on in the introduction the meaning of “adopting technologies”. In general, it seems like most of concepts here are given for granted but a clear specification is needed.

The PubMed database was added to the literature review in order to improve the scope of the research. In addition, "technology adoption" was better described in the introduction as requested by the reviewer.

5) Results. Please check if the number of articles excluded for missing data is correct since it’s more than half of collected paper and seems strange. Figure 3: please write the full name of journals for publication in respect to MDPI standards. Figure 5: please try to upload a higher quality image since without zooming is very difficult to read. Figure 6: I think that the introduction about SDGs should be pointed out in the introduction Section not here in the results. Here you also introduced AHP, please in the introduction cite it as one of the direct methods to elicit category weights. Please make a more detailed categorization of technologies applied to buildings starting from Figure 7. In particular, at least try to link specific technology to one or more of the individuated categories. Future trends and research gaps: please try to focus more on the gaps since as it is this section seems to be a collection of new technologies applied for IEQ evaluation.

  • Figure 3 became Figure 2 and, as requested, the full names of the journals were written to comply with MDPI standards.
  • Figure 5 became Figure 4 and was remodeled so that a new saving format could be applied to increase the quality and resolution of the image. With regard to the size of the letters, ScienceScape doesn't allow you to enlarge the letter as the site automatically generates the image when you upload the data extracted from the databases in Bibtex format, so zooming in is necessary.
  • Figure 6 was removed because we believed it took up considerable space in the article, so we removed it and left only the general definition in the introduction. As requested, the AHP method has been inserted in the introduction. Figure 7 became Table 2 so that a more detailed categorization of the technologies applied could be made. This categorization was based on the profile of each tool.

6) Conclusion: try to summarize the findings of the three research questions into a bullet list. In addition, future research to fill the individuated gaps should be proposed since is one of the scopes of a literature review to guide researchers in their future works.

The conclusions were listed as requested and some future research was suggested individually to fill in the gaps.

Reviewer 2 Report

Dear authors,

I wanted to reach out and express my sincerest appreciation for your efforts in writing this paper as I find it to be well-executed. Still I have some comments regarding some aspects of it.

The study scope is about the growing concern for sustainability, particularly in the context of ensuring resources for future generations and addressing the rising energy consumption in buildings. It highlights the importance of investigating IEQ, which encompasses factors like indoor air quality, temperature, noise, and lighting. It states that improving IEQ can have several positive impacts, including enhancing quality of life, reducing stress, saving energy, and promoting health, well-being, and productivity. Additionally, provides insights into how technological advancements and challenges in managing IEQ relate to the Sustainable Development Goals.

The content of this paper is complex and presents a tangible contribution by relating IEQ and SDGs, adding a valid contribution to the subject. Therefore, I recommend its publication but with some adjustments, that I consider to be "minor revisions".

Abbreviations in the Abstract: It's advisable to refrain from using abbreviations in the abstract, such as "IEQ" or "SDGs."

Concept of Bioclimatic Architecture: I believe it's crucial to contextualize the development and emergence of the concept of bioclimatic architecture within the content of lines 45 to 53. This concept (re)emerged as a response to the high energy and maintenance costs associated with HVAC systems and as a means to achieve a balanced Indoor Environmental Quality (IEQ) while keeping energy costs low.

Figures: The figures are generally appropriate but could benefit from some improvements. Certain figures, such as Figure 1 and 7, require resizing. Figure 2 might be better replaced with a table for improved quality. Figures like Figure 3 and 5 could be moved to appendices to enhance readability, as the information they contain is too small to be properly read.

Figure 3 should include a legend. While the meaning of bars and lines is explained in lines 161-164, details about colors and the different squares in colors should be clarified. Are these related to the year of publication or an interval?

Figure 7: Given the quality of work in the other figures, Figure 7 needs further enhancement. It should be more visually appealing and not appear as a low-quality screenshot.

Methodology: You mentioned identifying 475 articles in the Scopus database, with 151 remaining after applying exclusion criteria (as shown in Figure 1). Please refer to Figure 1 again to guide the reader even in further areas of the paper. Explain how StArt was used in this process and provide details about the inputs.

Clarify the role of ScienceScape in this analysis, as it was mentioned as a software used.

Conclusions: Define "GSAS" to avoid confusion. Pay careful attention to the use of abbreviations, as there are instances where "IEQ" and "SDG’s" are abbreviated multiple times, which can be confusing.

The conclusion section could benefit from further development. Consider expanding it to offer recommendations for future research directions in this field. Connect this with the response to RQ3 to provide guidance on what the next steps and recommendations for future research might be. In other words,  what were the research gaps and opportunities you identified.

Keep up the good work and all the best!

Dear Editor,

I found the english level perfectly acceptable with some minor spelling corrections here and there.

Kind Regards,

Author Response

Reviewer 2:

Dear reviewer, thank you very much for your time and your comments in order to improve this paper. Please see below the answers to all your comments.

Comments:

1) Abbreviations in the Abstract: It's advisable to refrain from using abbreviations in the abstract, such as "IEQ" or "SDGs."

The abbreviations in the summary have been removed as requested.

2) Concept of Bioclimatic Architecture: I believe it's crucial to contextualize the development and emergence of the concept of bioclimatic architecture within the content of lines 45 to 53. This concept (re)emerged as a response to the high energy and maintenance costs associated with HVAC systems and as a means to achieve a balanced Indoor Environmental Quality (IEQ) while keeping energy costs low.

The concept of Bioclimatic Architecture was inserted into the text as requested.

3) Figures: The figures are generally appropriate but could benefit from some improvements. Certain figures, such as Figure 1 and 7, require resizing. Figure 2 might be better replaced with a table for improved quality. Figures like Figure 3 and 5 could be moved to appendices to enhance readability, as the information they contain is too small to be properly read.

Figure 3 should include a legend. While the meaning of bars and lines is explained in lines 161-164, details about colors and the different squares in colors should be clarified. Are these related to the year of publication or an interval?

Figure 7: Given the quality of work in the other figures, Figure 7 needs further enhancement. It should be more visually appealing and not appear as a low-quality screenshot.

Some changes were made to the figures:

  • Figure 1 became Table 1 and immediately afterward the purposes of applying the established criteria were explained and Figure 7 became Table 2, in addition, the objectives of each of the categories were put;
  • With regard to Figure 2, it was suggested that it be replaced by a table, but it should be emphasized that the design of this figure is specific to this PRISMA methodology, so it doesn't detract too much from its purpose.
  • Figure 3 already has a legend at the bottom with the colors that represent the years of the publications, and the colors and the different stacked color squares have been explained in a better way;
  • Figures 3 and 5 have not been moved to the appendix because trying to place them in landscape format affected the document's formatting, especially in the page header. As we were unable to leave it in the best layout, we opted to leave it in its current position.
  • Figure 5 became Figure 4 due to the removal of some figures that we considered unnecessary. Regarding the resolution, the letters have become smaller because the figure is generated automatically on the ScienceScape website and when the data extracted from the databases is uploaded, the AKJ Sankey diagram is generated, which does not allow us to change the colors or size of the letters.
  • As Figure 7 was not of good quality, we decided to transform it into Table 2, so that we could list all the categories and explain the purpose of each one.

4) Methodology: You mentioned identifying 475 articles in the Scopus database, with 151 remaining after applying exclusion criteria (as shown in Figure 1). Please refer to Figure 1 again to guide the reader even in further areas of the paper. Explain how StArt was used in this process and provide details about the inputs.

Clarify the role of ScienceScape in this analysis, as it was mentioned as a software used.

Both StArt and ScienceScape were explained in more detail in the methodology, providing further details on the subject.

5) Conclusions: Define "GSAS" to avoid confusion. Pay careful attention to the use of abbreviations, as there are instances where "IEQ" and "SDG’s" are abbreviated multiple times, which can be confusing.

The conclusion section could benefit from further development. Consider expanding it to offer recommendations for future research directions in this field. Connect this with the response to RQ3 to provide guidance on what the next steps and recommendations for future research might be. In other words, what were the research gaps and opportunities you identified.

In the conclusion, "GSAS" was defined and some recommendations were made for future research based on the results found throughout the review.

Reviewer 3 Report

This paper conducts a systematic literature to explore the technological advancements and challenges in managing healthy and sustainable built environments, focusing on the interaction between Indoor Environmental Quality (IEQ) and Sustainable Development Goals (SDGs). The paper addresses an interesting topic with much room in the literature, especially with respect to the exploration of the interactions between IEQ and SDGs.

The quality of the paper could be significantly improved by implementing a series of changes. First, the research gap that the study attempts to cover must be more comprehensively described, alongside the rationale behind establishing these particular research questions. Furthermore, the results must be presented in a more coherent and specific way, to explicitly address the research questions set in the paper in a way that facilitates readers to grasp a solid understanding of the key findings. In this respect, I propose a reconsideration of the paper after major revision, in order the authors to bring it into a suitable form for publication in the ’’Urban Science’’ paper.

Below, I provide my detailed comments to which authors should pay attention.

i) The research gap in the ‘’Introduction’’ section must be better and more extensively explained, together with how this study intends to fill it. In this context, the relevant literature to the study's context should be discussed more extensively. This can also help balance the paper's structure, as currently a disproportionate part of the text is dedicated to results discussion and presentation, compared to the background information and the description of materials and methods.

ii) Furthermore, the research questions established in the paper come unexpectedly in Section 2, without explicitly explaining how they relate to the identified research gap. Research questions could better fit in the ‘’Introduction’’ section, where it should be explained how they relate to the identified research gap. The following structure could be adopted: first, identify the research gap and then explain how the study aims to address it by answering these research questions. Additionally, it must be ensured that each research question aligns with its respective objectives—for example in the 3rd research question, ''opportunities'' are mentioned only in the research question but not in the objectives. Also, the 2nd research question can be described more clearly to clarify how it differs from the 1st research question.

iii) The inclusion and exclusion criteria for the literature review, apart from Figure 1, must be described in the text. Furthermore, the 2nd inclusion criterion isn’t really a criterion, as it just mentions that there isn’t a restriction from the aspect of time, and as such, it may be removed. Also, the 4th inclusion criterion should become more specific. This could be done by adding one inclusion criterion for each of the three research questions, without conducting again the literature review. Moreover, the authors should clarify the meaning of the 6th exclusion criterion—Does it refer to the unavailability of full-text articles? Additionally, in the 8th exclusion criterion, it should be explained why book chapters and conference papers are excluded from the analysis. Finally, the text describing the 7th exclusion criterion should be grammatically corrected—this applies also to Figure 2 (‘’Articles only IEQ or SDQ’’).

iv) In section 4, the presentation of results could be improved by avoiding excessive text. Apart from making the paper’s structure more balanced (see also my comment above), this would make it easier for readers to grasp the key findings of the analysis, especially those that explicitly address the specific research questions. The authors must, at least, explicitly highlight the key outcomes in each research question in a coherent form, such as by presenting them in bullet points or tables. Additionally, the key results of the analysis, such as those mentioned in the abstract of the paper, need to be more profound so that it would be impossible to know them before performing the literature review. This applies mainly to the first outcome mentioned in the abstract, which could be declared a priori without performing a literature review.

The text is generally well-written, with only a few minor changes in certain areas to be required.

Author Response

Reviewer 3:

Dear reviewer, thank you very much for your time and your comments in order to improve this paper. Please see below the answers to all your comments.

Comments:

1) The research gap in the ‘’Introduction’’ section must be better and more extensively explained, together with how this study intends to fill it. In this context, the relevant literature to the study's context should be discussed more extensively. This can also help balance the paper's structure, as currently a disproportionate part of the text is dedicated to results discussion and presentation, compared to the background information and the description of materials and methods.

As requested, the research gap was better described in the introduction and explained more extensively and how the study can fill it. Furthermore, in order to balance the structure of the article, the research questions were included in the introduction, so that they could be directly linked to the research gap presented and explored throughout the study.

2) Furthermore, the research questions established in the paper come unexpectedly in Section 2, without explicitly explaining how they relate to the identified research gap. Research questions could better fit in the ‘’Introduction’’ section, where it should be explained how they relate to the identified research gap. The following structure could be adopted: first, identify the research gap and then explain how the study aims to address it by answering these research questions. Additionally, it must be ensured that each research question aligns with its respective objectives—for example in the 3rd research question, ''opportunities'' are mentioned only in the research question but not in the objectives. Also, the 2nd research question can be described more clearly to clarify how it differs from the 1st research question.

The research questions were moved to the introduction section, bringing a greater connection to the theme proposed in the article through the research gap addressed, thus bringing greater alignment with the research objective.

3) The inclusion and exclusion criteria for the literature review, apart from Figure 1, must be described in the text. Furthermore, the 2nd inclusion criterion isn’t really a criterion, as it just mentions that there isn’t a restriction from the aspect of time, and as such, it may be removed. Also, the 4th inclusion criterion should become more specific. This could be done by adding one inclusion criterion for each of the three research questions, without conducting again the literature review. Moreover, the authors should clarify the meaning of the 6th exclusion criterion—Does it refer to the unavailability of full-text articles? Additionally, in the 8th exclusion criterion, it should be explained why book chapters and conference papers are excluded from the analysis. Finally, the text describing the 7th exclusion criterion should be grammatically corrected—this applies also to Figure 2 (‘’Articles only IEQ or SDQ’’).

The inclusion and exclusion criteria have been placed in Table 1 to optimize the space of the article and explained more consistently in the paragraph following this table. In addition, figure 2, which corresponds to PRISMA, has been grammatically tidied up as requested and another database, PUBMED, has been added to improve the answers to the RQs.

4) In section 4, the presentation of results could be improved by avoiding excessive text. Apart from making the paper’s structure more balanced (see also my comment above), this would make it easier for readers to grasp the key findings of the analysis, especially those that explicitly address the specific research questions. The authors must, at least, explicitly highlight the key outcomes in each research question in a coherent form, such as by presenting them in bullet points or tables. Additionally, the key results of the analysis, such as those mentioned in the abstract of the paper, need to be more profound so that it would be impossible to know them before performing the literature review. This applies mainly to the first outcome mentioned in the abstract, which could be declared a priori without performing a literature review.

Section 4, in particular 4.1, has been split into three subsections to make the structure of the article more balanced as requested and the summary has also been improved.

Reviewer 3:

Dear reviewer, thank you very much for your time and your comments in order to improve this paper. Please see below the answers to all your comments.

Comments:

1) The research gap in the ‘’Introduction’’ section must be better and more extensively explained, together with how this study intends to fill it. In this context, the relevant literature to the study's context should be discussed more extensively. This can also help balance the paper's structure, as currently a disproportionate part of the text is dedicated to results discussion and presentation, compared to the background information and the description of materials and methods.

As requested, the research gap was better described in the introduction and explained more extensively and how the study can fill it. Furthermore, in order to balance the structure of the article, the research questions were included in the introduction, so that they could be directly linked to the research gap presented and explored throughout the study.

2) Furthermore, the research questions established in the paper come unexpectedly in Section 2, without explicitly explaining how they relate to the identified research gap. Research questions could better fit in the ‘’Introduction’’ section, where it should be explained how they relate to the identified research gap. The following structure could be adopted: first, identify the research gap and then explain how the study aims to address it by answering these research questions. Additionally, it must be ensured that each research question aligns with its respective objectives—for example in the 3rd research question, ''opportunities'' are mentioned only in the research question but not in the objectives. Also, the 2nd research question can be described more clearly to clarify how it differs from the 1st research question.

The research questions were moved to the introduction section, bringing a greater connection to the theme proposed in the article through the research gap addressed, thus bringing greater alignment with the research objective.

3) The inclusion and exclusion criteria for the literature review, apart from Figure 1, must be described in the text. Furthermore, the 2nd inclusion criterion isn’t really a criterion, as it just mentions that there isn’t a restriction from the aspect of time, and as such, it may be removed. Also, the 4th inclusion criterion should become more specific. This could be done by adding one inclusion criterion for each of the three research questions, without conducting again the literature review. Moreover, the authors should clarify the meaning of the 6th exclusion criterion—Does it refer to the unavailability of full-text articles? Additionally, in the 8th exclusion criterion, it should be explained why book chapters and conference papers are excluded from the analysis. Finally, the text describing the 7th exclusion criterion should be grammatically corrected—this applies also to Figure 2 (‘’Articles only IEQ or SDQ’’).

The inclusion and exclusion criteria have been placed in Table 1 to optimize the space of the article and explained more consistently in the paragraph following this table. In addition, figure 2, which corresponds to PRISMA, has been grammatically tidied up as requested and another database, PUBMED, has been added to improve the answers to the RQs.

4) In section 4, the presentation of results could be improved by avoiding excessive text. Apart from making the paper’s structure more balanced (see also my comment above), this would make it easier for readers to grasp the key findings of the analysis, especially those that explicitly address the specific research questions. The authors must, at least, explicitly highlight the key outcomes in each research question in a coherent form, such as by presenting them in bullet points or tables. Additionally, the key results of the analysis, such as those mentioned in the abstract of the paper, need to be more profound so that it would be impossible to know them before performing the literature review. This applies mainly to the first outcome mentioned in the abstract, which could be declared a priori without performing a literature review.

Section 4, in particular 4.1, has been split into three subsections to make the structure of the article more balanced as requested and the summary has also been improved.

Reviewer 3:

Dear reviewer, thank you very much for your time and your comments in order to improve this paper. Please see below the answers to all your comments.

Comments:

1) The research gap in the ‘’Introduction’’ section must be better and more extensively explained, together with how this study intends to fill it. In this context, the relevant literature to the study's context should be discussed more extensively. This can also help balance the paper's structure, as currently a disproportionate part of the text is dedicated to results discussion and presentation, compared to the background information and the description of materials and methods.

As requested, the research gap was better described in the introduction and explained more extensively and how the study can fill it. Furthermore, in order to balance the structure of the article, the research questions were included in the introduction, so that they could be directly linked to the research gap presented and explored throughout the study.

2) Furthermore, the research questions established in the paper come unexpectedly in Section 2, without explicitly explaining how they relate to the identified research gap. Research questions could better fit in the ‘’Introduction’’ section, where it should be explained how they relate to the identified research gap. The following structure could be adopted: first, identify the research gap and then explain how the study aims to address it by answering these research questions. Additionally, it must be ensured that each research question aligns with its respective objectives—for example in the 3rd research question, ''opportunities'' are mentioned only in the research question but not in the objectives. Also, the 2nd research question can be described more clearly to clarify how it differs from the 1st research question.

The research questions were moved to the introduction section, bringing a greater connection to the theme proposed in the article through the research gap addressed, thus bringing greater alignment with the research objective.

3) The inclusion and exclusion criteria for the literature review, apart from Figure 1, must be described in the text. Furthermore, the 2nd inclusion criterion isn’t really a criterion, as it just mentions that there isn’t a restriction from the aspect of time, and as such, it may be removed. Also, the 4th inclusion criterion should become more specific. This could be done by adding one inclusion criterion for each of the three research questions, without conducting again the literature review. Moreover, the authors should clarify the meaning of the 6th exclusion criterion—Does it refer to the unavailability of full-text articles? Additionally, in the 8th exclusion criterion, it should be explained why book chapters and conference papers are excluded from the analysis. Finally, the text describing the 7th exclusion criterion should be grammatically corrected—this applies also to Figure 2 (‘’Articles only IEQ or SDQ’’).

The inclusion and exclusion criteria have been placed in Table 1 to optimize the space of the article and explained more consistently in the paragraph following this table. In addition, figure 2, which corresponds to PRISMA, has been grammatically tidied up as requested and another database, PUBMED, has been added to improve the answers to the RQs.

4) In section 4, the presentation of results could be improved by avoiding excessive text. Apart from making the paper’s structure more balanced (see also my comment above), this would make it easier for readers to grasp the key findings of the analysis, especially those that explicitly address the specific research questions. The authors must, at least, explicitly highlight the key outcomes in each research question in a coherent form, such as by presenting them in bullet points or tables. Additionally, the key results of the analysis, such as those mentioned in the abstract of the paper, need to be more profound so that it would be impossible to know them before performing the literature review. This applies mainly to the first outcome mentioned in the abstract, which could be declared a priori without performing a literature review.

Section 4, in particular 4.1, has been split into three subsections to make the structure of the article more balanced as requested and the summary has also been improved.

Round 2

Reviewer 1 Report

Paper is now acceptable for publication. Thank you for your effort that enached the paper quality and readability.

English is fine, only minor proofreading before publication is needed. 

Reviewer 3 Report

The authors have addressed my comments and therefore I recommend accepting the paper for publication in Urban Science.